# Influence of Synthesis Conditions on Gadolinium-Substituted Tricalcium Phosphate Ceramics and Its Physicochemical, Biological, and Antibacterial Properties

**DOI:** 10.3390/nano12050852

**Published:** 2022-03-03

**Authors:** Inna V. Fadeeva, Dina V. Deyneko, Katia Barbaro, Galina A. Davydova, Margarita A. Sadovnikova, Fadis F. Murzakhanov, Alexander S. Fomin, Viktoriya G. Yankova, Iulian V. Antoniac, Sergey M. Barinov, Bogdan I. Lazoryak, Julietta V. Rau

**Affiliations:** 1A.A. Baikov Institute of Metallurgy and Material Science, Russian Academy of Sciences, Leninsky Prospect 49, 119334 Moscow, Russia; fadeeva_inna@mail.ru (I.V.F.); alex_f81@mail.ru (A.S.F.); barinov_s@mail.ru (S.M.B.); 2Chemistry Department, Lomonosov Moscow State University, 119991 Moscow, Russia; deynekomsu@gmail.com (D.V.D.); bilazoryak@gmail.com (B.I.L.); 3Laboratory of Arctic Mineralogy and Material Sciences, Kola Science Centre, Russian Academy of Sciences, 14 Fersman Str., 184209 Apatity, Russia; 4Istituto Zooprofilattico Sperimentale Lazio e Toscana “M. Aleandri”, Via Appia Nuova, 00178 Rome, Italy; katia.barbaro@izslt.it; 5Institute of Theoretical and Experimental Biophysics, Russian Academy of Sciences, Institutskaya 3, 142290 Moscow, Russia; davidova_g@mail.ru; 6Institute of Physics, Kazan Federal University, Kremlevskaya 18, 420008 Kazan, Russia; margaritaasadov@gmail.com (M.A.S.); murzakhanov.fadis@yandex.ru (F.F.M.); 7Institute of Pharmacy, Department of Analytical, Physical and Colloid Chemistry, I.M. Sechenov First Moscow State Medical University, 8 Trubetskaya Street, Build. 2, 119991 Moscow, Russia; yankova_v_g@staff.sechenov.ru; 8Faculty of Materials Science and Engineering, University Politehnica of Bucharest, 313 Splaiul Independentei Street, District 6, 060042 Bucharest, Romania; antoniac.iulian@gmail.com; 9Istituto di Struttura Della Materia, Consiglio Nazionale Delle Ricerche (ISM-CNR), Via del Fosso del Cavaliere 100, 00133 Rome, Italy

**Keywords:** gadolinium, tricalcium phosphate, gadolinium-substituted tricalcium phosphate, gadolinium tricalcium phosphate powder, gadolinium tricalcium phosphate ceramics, gadolinium doped tricalcium phosphate

## Abstract

Gadolinium-containing calcium phosphates are promising contrast agents for various bioimaging modalities. Gadolinium-substituted tricalcium phosphate (TCP) powders with 0.51 wt% of gadolinium (0.01Gd-TCP) and 5.06 wt% of (0.1Gd-TCP) were synthesized by two methods: precipitation from aqueous solutions of salts (1) (Gd-TCP-pc) and mechano-chemical activation (2) (Gd-TCP-ma). The phase composition of the product depends on the synthesis method. The product of synthesis (1) was composed of *β*-TCP (main phase, 96%), apatite/chlorapatite (2%), and calcium pyrophosphate (2%), after heat treatment at 900 °C. The product of synthesis (2) was represented by *β*-TCP (main phase, 73%), apatite/chlorapatite (20%), and calcium pyrophosphate (7%), after heat treatment at 900 °C. The substitution of Ca^2+^ ions by Gd^3+^ in both *β*-TCP (main phase) and apatite (admixture) phases was proved by the electron paramagnetic resonance technique. The thermal stability and specific surface area of the Gd-TCP powders synthesized by two methods were significantly different. The method of synthesis also influenced the size and morphology of the prepared Gd-TCP powders. In the case of synthesis route (1), powders with particle sizes of tens of nanometers were obtained, while in the case of synthesis (2), the particle size was hundreds of nanometers, as revealed by transmission electron microscopy. The Gd-TCP ceramics microstructure investigated by scanning electron microscopy was different depending on the synthesis route. In the case of (1), ceramics with grains of 1–50 μm, pore sizes of 1–10 µm, and a bending strength of about 30 MPa were obtained; in the case of (2), the ceramics grain size was 0.4–1.4 μm, the pore size was 2 µm, and a bending strength of about 39 MPa was prepared. The antimicrobial activity of powders was tested for four bacteria (*S. aureus*, *E. coli*, *S. typhimurium*, and *E. faecalis*) and one fungus (*C**. albicans*), and there was roughly 30% of inhibition of the micro-organism’s growth. The metabolic activity of the NCTC L929 cell and viability of the human dental pulp stem cell study demonstrated the absence of toxic effects for all the prepared ceramic materials doped with Gd ions, with no difference for the synthesis route.

## 1. Introduction

Nowadays, a rapid increase in the aging of the population has been observed. Worldwide, about 20% of the population (around 2 billion) is foreseen to be over 60 years old by 2050 [1]. According to the World Health Organization, around 40% of people over 60 years old experience musculoskeletal disorders [1,2]. The aging of a population is characterized by a progressive bone loss of individuals. Osteopenia is associated with several outcomes, including functional decline, high risk of fractures, and frailty. Musculoskeletal disorders are among the key causes of morbidity in elderly people. For all these reasons, there is an enormous demand for synthetic bone substitutes and new tissue regeneration strategies. Various synthetic bone grafts have been developed over the past few decades, a part of them being based on calcium phosphates similar to those present in the human body, such as hydroxyapatite (HA, Ca_10_(PO_4_)_6_(OH)_2_)) and tricalcium phosphate (*β*-TCP, *β*-Ca_3_(PO_4_)_2_) [3,4,5].

*β*-TCP is one of the most used and promising synthetic bone graft substitutes, being osteoconductive and osteoinductive. Such properties combined with resorption mediated by cells result in the complete bone defect regeneration [4]. *β*-TCP has a higher solubility and, accordingly, rate of resorption in comparison with HA. In the case of *β*-TCP, the period of replacement of the majority of artificial material with natural bone is about 6–24 months [6,7,8]. It is possible to regulate the rate of resorption, physical features, and biological performance by cationic or anionic substitutions in the *β*-TCP structure [9,10,11]. For example, doping the *β*-TCP structure with various metal cations, such as copper (II), manganese (II), iron (III), and zinc (II) [12,13,14,15], leads to its destabilization and, consequently, to the increase in the rate of dissolution of these cations in the body fluid. In addition to the modulation of physicochemical characteristics of *β*-TCP, metal-substituted TCPs provide new advanced properties to the materials. Antibacterial characteristics can be imparted by Cu^2+^, Fe^3+^, Zn^2+^, and Ag^+^ ions [12,14,15,16], and the biocompatibility properties and, namely, cell proliferation on the materials surface can be improved by Fe^3+^ and Sr^2+^ ions substitution [5,14,17,18,19]. Mg-doped *β*-TCP biomaterials are able to enhance the cell proliferation and viability of human bone marrow-derived mesenchymal stem cells [20] and the differentiation of hBMSCs into osteoblasts [21].

For tissue regeneration, each step of bone healing needs to be optimized followed by imaging techniques, such as X-ray radiography, computed tomography (CT), single-photon emission computerized tomography (SPECT), and magnetic resonance imaging (MRI). However, even when using these methods, it is difficult to follow the processes of bone tissue regeneration and to control the degradation of synthetic bone substitutes without any contrast-providing substances. This problem can be solved by adding contrast agents to *β*-TCP bone substitute materials [22]. Calcium phosphate nanoparticles were employed as contrast agents by doping with europium (Eu^3+^), gadolinium (Gd^3+^), iron (Fe^3+^), neodymium (Nd^3+^), terbium (Tb^3+^), dysprosium (Dy^3+^), etc., for such imaging modalities as fluorescence, CT, MR, and nuclear imaging [23]. The synthesis of Gd^3+^-doped HA for a dual-modal probe for SPECT and MRI was developed in [24,25]. Gd-doped nanocomposites with enhanced MRI and X-ray dual imaging were developed in [26] to simulate the electrical properties of bone.

A number of studies have been devoted to Gd^3+^-substituted *β*-TCP materials [27,28]. It was demonstrated that Gd^3+^-doped *β*-Ca_3_(PO_4_)_2_ exhibited nontoxic behavior toward MG-63 cells and paramagnetic behavior [27]. In [28], the Gd, Ce, and Tb co-doped *β*-TCP porous nanospheres exhibited excellent cytocompatibility and were found to be promising for bioimaging guidance. 

The present study was aimed at the investigation of the influence of two synthesis routes of the Gd-substituted *β*-TCP: precipitation from aqueous solutions of salts [9] (1) and mechano-chemical activation (2), and of gadolinium concentration on the phase composition, morphology, and sintering behavior of Gd-TCP powders, and the microstructure and properties of the corresponding Gd-substituted tricalcium phosphate ceramics for possible use as contrast agents in bioimaging techniques. The prepared powder samples were characterized by the Fourier-transform infrared spectroscopy (FT-IR), X-ray diffraction (XRD), electron paramagnetic resonance spectroscopy (EPR), thermogravimetric analysis (TGA), specific surface area, scanning electrons microscopy (SEM), and transmission electron microscopy (TEM) methods. Antimicrobial in vitro tests using four bacteria (*S. aureus, E. coli, S. typhimurium**,* and *E. faecalis*) and one fungus (*C. albicans*) were performed. The fibroblast cell line NCTC clone L929 was used for cytotoxicity studies, and the viability of postnatal human dental pulp stem cells (DPSCs) was investigated.

## 2. Materials and Methods

Gadolinium-substituted tricalcium phosphates with 0.51 wt% (0.01Gd-TCP) and 5.06 wt% (0.1Gd-TCP) of gadolinium were synthesized by two methods: precipitation from aqueous solutions of salts (1) [9] and mechano-chemical activation (2) [29]. 

The raw materials for precipitation from aqueous solutions of salts were calcium nitrate (chemical grade, Chimmed, Moscow, Russia), diammonium phosphate (analytical grade, Chimmed, Moscow, Russia), gadolinium chloride (chemical grade, Chimmed, Moscow, Russia), and 25% aqueous ammonia solution (analytical grade, Chimmed, Moscow, Russia). Synthesis of Gd-substituted TCP was performed according to reactions (1) and (2):2.9 Ca(NO_3_)_2_ + 0.067 GdCl_3_ + 2 (NH_4_)_2_HPO_4_ + 2 NH_4_OH → Ca_2.9_Gd_0.067_(PO_4_)_2_ + 0.2 NH_4_Cl + 5.8 NH_4_NO_3_ + 2 H_2_O(1)
2.99 Ca(NO_3_)_2_ + 0.0067 GdCl_3_ + 2 (NH_4_)_2_HPO_4_ + 2 NH_4_OH → Ca_2.99_Gd_0.0067_(PO_4_)_2_ + 0.02 NH_4_Cl + 5.98 NH_4_NO_3_ + 2 H_2_O(2)

The samples obtained by the precipitation method were named 0.1-Gd-TCP-pc (Reaction (1)) and 0.01Gd-TCP-pc (Reaction (2)).

The mechano-chemical activation was carried out in the planetary mill container PM-1, (Vibrotechnik, St. Petersburg, Russia) at a 1500 min^–1^ rotation rate. The ratio between materials mixture: zirconium oxide balls was 1:5 and the activation time was 30 min. The amounts of the reagents were calculated according to Reactions (3) and (4):2.9 CaO + 0.067 GdCl_3_ + 2 (NH_4_)_2_HPO_4_ → Ca_2.9_Gd_0.067_(PO_4_)_2_ + 3.8 NH_3_ + 0.2 NH_4_Cl + 2.9 H_2_O(3)
2.99 CaO + 0.0067 GdCl_3_ + 2 (NH_4_)_2_HPO_4_ → Ca_2.9_Gd_0.0067_(PO_4_)_2_ + 3.8 NH_3_ + 0.02 NH_4_Cl + 2.9 H_2_O(4)

Immediately after the synthesis, 200 mL of water was added to the reagents, and grinding was continued for 30 min. The precipitate was filtered out using the Buchner funnel and dried at 110 °C for 12–15 h. The samples obtained by the mechano-chemical activation method were named 0.1Gd-TCP-ma (Reaction (3)) and 0.01Gd-TCP-ma (Reaction (4)).

The powders obtained by synthesis methods (1) and (2) were heat-treated at 900 °C for 1 h. The corresponding ceramics were prepared by uniaxial pressing of the powders in a steel mold under pressure of 100 MPa, followed by the sintering of compacts in air atmosphere in the muffle furnace at 1100 °C for 2 h.

Fourier-transform infrared spectroscopy absorption spectra of the synthesized powders mixed with KBr were recorded in the range of 400–4000 cm^–1^ with a resolution of 0.9 cm^–1^ by means of the Thermo Nicolet Avatar 330 FTIR spectrometer (Thermo Fisher Scientific, Waltham, MA, USA).

Powder X-ray diffraction patterns were collected on a Rigaku D/MAX 2500 (Ni-radiation filter Cu Kα, θ/2θ geometry). XRD data were obtained at room temperature in the 2θ range between 10° and 60° with a step interval of 0.02° and 3°/min scan rate. The LeBail decomposition [30] was applied using the JANA2006 software for unit cell parameters and volume determination. Phase analysis was carried out by means of the Crystallographica Search-March program (version 2.0.3.1) and the JCPDS PDF-2 and PDF-4 databases. The Rietveld method was applied for quantitative phase analysis using the JANA2006 software [31]. Crystallographic data of the space group (SG), unit cell, and atomic coordinates of *β*-Ca_3_(PO_4_)_2_ (PDF#4 No. 70-2065) [32], Ca_10_(PO_4_)_6_(OH)_2_ (ICDD 183744) [33], and Ca_2_P_2_O_7_ (PDF#4 No. 04-009-3876) [34] were applied as initial parameters. The fifteenth-order polynomial was applied to refine the background and modified pseudo-Voigt function-peak profiles. The unit cell parameters were refined, but the atomic coordinates were taken without refinement. After the last refinement procedure, a good agreement was found between the experimental and calculated patterns.

Electron paramagnetic resonance spectra in pulse mode were recorded using a Bruker Elexsys E580 spectrometer in the X-band microwave range (ν_MW_ = 9.61 GHz) at the temperatures of 12 and 25 K. The duration of the microwave π/2 pulse in the pulse sequences was 16 ns. The technique for detecting the integral intensity of the electron spin echo during the sweep of the magnetic field B_0_ was used to measure the EPR spectra. The relaxation processes were measured using a Hahn sequence with a time variation between the first and second pulses from 180 to 4276 ns. The electron-nuclear interaction spectra were recorded using the HYSCORE sequence (hyperfine sublevel correlation) with a change in both distances (τ and T) from 180 to 1204 ns. Gadolinium-substituted TCP-0.001Gd-TCP with a gadolinium content of 0.0067 wt% was synthesized for the EPR spectroscopic studies by the precipitation method.

Thermogravimetric analysis was carried out by a thermogravimetric analyzer Model: Q-1500 D (Paulik and Erdei, Hungary) in the temperature range of 20–1200 °C with a heating rate of 5 °/min.

The specific surface area of powders was measured by low-temperature nitrogen adsorption by the Brunauer–Emmett–Teller (BET) method. The samples were preheated in vacuum to remove the adsorbed water. Afterward, the samples placed in an ampoule filled with helium were immersed in a Dewar vessel filled with liquid nitrogen. The ampoule was evacuated and cooled to a temperature of 77 K. The external pressure corresponded to the pressure of saturated nitrogen vapor at T = 77 K. As a next step, nitrogen gas was dosed until the desired pressure was reached in the ampoule (0.30 atm), and the volume of injected nitrogen was measured. 

Scanning electron microscopy observations of Gd-substituted TCP were performed to study the microstructure of the ceramic materials by a Tescan Vega II scanning electron microscope. The grain size was calculated by the random secant method from SEM data. The method consists of calculating the intersection of grain boundaries of a random secant, which is the middle line of the eyepiece–micrometer. This method determines the average diameter in the case of equiaxial grains.

Transmission electron microscopy images of pure *β*-TCP-pc, *β*-TCP-ma, and Gd-substituted TCP—0.1Gd-TCP-pc and 0.01Gd-TCP-pc—were obtained by using a LEO 912 AB OMEGA microscope (Carl Zeiss, Oberckochen, Germany). 

The antimicrobial activity was evaluated for 0.1Gd-TCP-pc and 0.1Gd-TCP-ma powder samples using four bacteria (*S. aureus*, *E. coli*, *S. typhimurium*, and *E. faecalis*) and one fungus (*C. albicans*). First, the tested samples were autoclaved at 121 °C for 20 min at a pressure of 1.1 bar and diluted to 1:1000 (*w/v*) in Brain Heart Infusion (BHI, DIFCO, Sparks, NV, USA). Each microorganism was grown in the presence of the tested powder samples for 24 h, under slow stirring. The temperature for the bacteria was 37 °C, while that for the fungus was 28 °C. The growth of microorganisms was assessed by reading the OD600_nm_ (optical density at 600 nm (λ adsorption)) of the growth medium with a Biophotometer D30 (Eppendorf, Hamburg, Germany). The experiments were carried out in triplicate, and the results are expressed as the mean value ± standard deviation (SD).

Cytotoxicity was investigated using extracts prepared according to the requirements of GOST R ISO 10993.5-2011 and GOST ISO 10993-12-2015 (Russian Federation) on fibroblast cell line NCTC clone L929 in the culture medium DMEM/F12 using the MTT test.

To study the viability of cells on the surface of ceramic materials, we used postnatal human dental pulp stem cells (DPSCs) isolated from molar rudiment extracted in accordance with the orthodontic indications [35], supported by the voluntary consent of donors. For investigations, the cell culture at the 5th passage was applied.

Sterilized samples were placed in the wells of a 24-well plate and DPSC cells were seeded at a concentration of 30,000 cells/cm^2^ in DMEM/F12 medium (1:1) supplemented with 10% fetal bovine serum (FBS), 100 U/mL penicillin/streptomycin, and 2 mL glutamine, and cultured at 37 °C in an atmosphere of 5% CO_2_.

Cell viability was assessed by differential fluorescent staining of living and dead cells using the SYTO 9 fluorescent dyes, propidium iodide and Hoechst 33342. Microphotographies were carried out by means of an Axiovert 200 inverted microscope (Zeiss Oberckochen, Germany). In the modes λ_exc_ = 450–490 nm and λ_em_ = 515–565 nm, fluorescent dye SYTO 9 stained green DNA and RNA of living and dead cells. In the study modes λ_exc_ = 546 nm and λ_em_ = 575–640 nm, the intercalating reagent propidium iodide (PI) stained the nuclei of dead cells red. In the study modes λ_exc_ = 355 nm and λ_em_ = 460 nm, fluorescent dye Hoechst 33,342 stained DNA of living and dead cells blue.

The percentage of viable cells was calculated on a fixed surface area (in the micrographs) by the number of cell nuclei stained with Hoechst 33,342 (all cells) and PI (dead cells).

To obtain cell viability and survivability results, 8 repetitions were made for each point. The calculation of standard deviation was performed for each point in all the sections. The statistical analysis of the reliability was carried out according to the Mann–Whitney U-test (*p* ≤ 0.01).

## 3. Results

### 3.1. FT-IR Study

FT-IR spectra of the Gd-substituted TCP powders prepared by methods (1) and (2) are shown in Figure 1. The spectra are characterized by the distinctive features of the substituted *β*-TCP-type structure compounds (Figure 1). Assignment of the vibration bands from the FT-IR spectrum of Gd-TCP is provided in Table 1. According to the FT-IR results, the formation of the impurity Ca_2_P_2_O_7_ phase is confirmed, which is consistent with the XRD data shown in the next section. Vibrations of hydroxyl groups appear only in samples obtained by the mechano-chemical activation method (2). The position of the OH^–^ bands, the absence of a band at 633 cm^–1^, as well as the splitting of the vibrations of the hydroxyl group (3644 cm^–1^, 3544 cm^–1^) for the 0.1Gd-TCP-ma sample (Figure 1) may indicate that Cl^–^ ions enter the structure of the apatite phase. The presence of bands assigned to the carbonate group should also be noted, and it indicates partial replacement of the groups PO_4_^3–^ → CO_3_^2^^–^. It is known that the size of powder particles can also affect the shape of the FTIR spectra of samples [36]. However, according to the TEM data, the particle size of the powder obtained by wet precipitation is close to each other, and the differences in the spectra can be attributed to the content of Gd in the samples. Samples obtained by the mechano-chemical activation are multi-phase samples and contain the TCP phase, as well as the apatite phase. Therefore, it is difficult to isolate the effect of dispersion on the nature and shape of the IR spectra.

### 3.2. X-Ray Diffraction Study

Figure 2 shows the XRD patterns of 0.1 Gd-TCP samples obtained by two methods. According to the XRD data, *β*-TCP is the main crystalline phase in the synthesized samples, after calcination at 900 °C. In addition, two impurity phases are found: chlorapatite (Ca_10_(PO_4_)_6_Cl_2_, Cl-AP-phase) and *β*-calcium pyrophosphate (*β*-Ca_2_P_2_O_7_). The formation of the chlorapatite phase is associated with the use of GdCl_3_ as a precursor. Thus, the presence of Cl^–^ ions in the reaction mixture promotes the formation of a thermodynamically stable chlorapatite phase [33]. Ca_2_P_2_O_7_ is formed during the annealing from a by-product of the reaction of dicalcium phosphate dihydrate (CaHPO_4_·2H_2_O), according to Reaction (5):2 CaHPO_4_·2 H_2_O → Ca_2_P_2_O_7_ + 5 H_2_O(5)

The quantitative phase analysis of all the synthesized samples is shown in Table 2 along with the reference data for the pure TCP, chlorapatite, HA, and pyrophosphate phases. The phase composition of 0.1Gd-TCP-ma is the most inhomogeneous compared to other samples (Figure 2). The main phase is *β*-TCP (73 wt%), and the content of apatite is significant (20 wt%), compared to 0.1Gd-TCP-pc (2 wt%). The pyrophosphate phase is detected in all synthesized samples, with an average content in the range of 2–7%. The scheme of the substitution is 3Ca^2+^ → 2R^3+^ + Υ (Υ-vacancy), with the formation of a vacancy, and no additional charge compensators are required. The values of the ionic radii of calcium (*r*_VI_ = 1.00 Å, *r*_VIII_ = 1.12 Å) and gadolinium (*r*_VI_ = 0.94 Å, *r*_VIII_ = 1.05 Å) [37] ions in this substitution should have an impact on the unit cell parameters and volume. Previously [38], it was shown that Gd^3+^ ions occupy only *M*1–*M*3 sites in the *β*-TCP-type structure. There are no reported data on the unit cell changes in the Ca_3-*x*_Gd_2*x*/3_(PO_4_)_2_ solid solutions. However, such data are available for Ca_3-*x*_Eu_2*x*/3_(PO_4_)_2_ solid solutions [39]. When Ca^2+^ ions are replaced by Eu^3+^, the parameter *a* and the unit cell volume *V* increase. As the radii of Gd^3+^ (*r*_VIII_ = 1.05 Å) and Eu^3+^ (*r*_VIII_ = 1.07 Å) [37] slightly differ, it can be assumed that the unit cell parameters will change similarly when Ca^2+^ is replaced by Gd^3+^. If we assume that all the Gd^3+^ ions during the synthesis enter the *β*-TCP lattice, then the composition of this phase is described by the formula Ca_2.9_Gd_0.067_(PO_4_)_2_ (0.1Gd-TCP-pc). Comparing the unit cell parameters of Ca_2.9_Gd_0.067_(PO_4_)_2_, Ca_2.928_Eu_0.048_(PO_4_)_2_, and Ca_2.858_Eu_0.095_(PO_4_)_2_ with a similar composition (see Table 2), one can observe that most of the Gd^3+^ ions are included in the *β*-TCP structure upon its preparation by the precipitation method. In this case, the unit cell volume is noticeably larger for 0.1Gd-TCP-pc than for the pure *β*-TCP (Table 2). The cell volume of 0.1Gd-TCP-ma obtained by mechano-chemical activation is smaller than that of pure *β*-TCP (Table 2). These data indicate the existence of impurity amounts of Gd^3+^ in the 0.1Gd-TCP-ma sample.

The calculated unit cell parameters for the impurity apatite phase are noticeably smaller compared to the pure Cl-AP (apatite) phase (Table 2). Therefore, we made an assumption on the existence of a mixture of phases: Cl-AP and HA in the synthesized samples (this assumption is in agreement with the obtained FT-IR results). In the impurity HA phase, some Ca^2+^ ions might be substituted by the Gd^3+^ ions with the formation of oxyapatite according to (6) [40]:OH^−^ + Ca^2+^ → Gd^3+^ + O^2−^(6)

**Table 2 nanomaterials-12-00852-t002:** The results of the quantitative phase analysis for Ca_2.9_Gd_0.067_(PO_4_)_2_ (0.1Gd-TCP) and Ca_2.9_Gd_0.0067_(PO_4_)_2_ (0.01Gd-TCP) synthesized by precipitation and mechano-chemical activation methods.

	Phase, SG	wt% (JANA 2006)	*a*, Å	*c*, Å	*V*, Å^3^
0.1Gd-TCP-ma	*β*-TCP, *R*3*c*	73	10.423(6)	37.391(3)	3518.3(8)
Ca_2.9_Gd_0.067_(PO_4_)_2_	Cl-AP, *P*6_3_/*m*	20	9.435(8)	6.867(4)	529.5(8)
	Ca_2_P_2_O_7_, *P*4_1_	7	6.686(5)	24.148(6)	1081.0(4)
0.1Gd-TCP-pc	*β*-TCP, *R*3*c*	96	10.443(5)	37.438(9)	3536.3(8)
Ca_2.9_Gd_0.067_(PO_4_)_2_	Cl-AP, *P*6_3_/*m*	2	9.339(7)	6.918(6)	522.7(8)
	Ca_2_P_2_O_7_, *P*4_1_	2	6.684(2)	24.149(4)	1080.5(4)
0.01Gd-TCP-ma	*β*-TCP, *R*3*c*	87	10.437(4)	37.403(2)	3529.1(9)
Ca_2.9_Gd_0.0067_(PO_4_)_2_	Cl-AP, *P*6_3_/*m*	7			
	Ca_2_P_2_O_7_, *P*4_1_	6	6.685(9)	24.148(4)	1082.1(6)
0.01Gd-TCP-pc	*β*-TCP, *R*3*c*	96	10.438(1)	37.406(2)	3526.4(8)
Ca_2.9_Gd_0.0067_(PO_4_)_2_	Cl-AP, *P*6_3_/*m*	0	-	-	-
	Ca_2_P_2_O_7_, *P*4_1_	4			
*β*-Ca_3_(PO_4_)_2_ [32]	*R*3*c* (Z = 21)		10.435(2)	37.402(9)	3527.2(6)
Ca_2.928_Eu_0.048_(PO_4_)_2_ [39]	*R*3*c* (Z = 21)		10.440(1)	37.380(1)	3528.0(9)
Ca_2.858_Eu_0.095_(PO_4_)_2_ [39]	*R*3*c* (Z = 21)		10.440(1)	37.39(1)	3529
Ca_10_(PO_4_)_6_Cl_2_ [41]	*P*6_3_/*m* (Z = 1)		9.590(2)	6.766(6)	538.96
Ca_10_(PO_4_)_6_(OH)_2_ [42]	*P*6_3_/*m* (Z = 1)		9.4207(1)	6.8817(1)	528.92
(Ca_4.87_Gd_0.11_)_2_(PO_4_)_6_(OH)_2_ [43]	*P*6_3_/*m* (Z = 1)		9.41284(6)	6.8815(1)	528.03(9)
Ca_2_P_2_O_7_ [34]	*P*4_1_ (Z = 8)		6.6858(8)	24.147(5)	1081.1(3)

The decrease in the unit cell parameters and volume is a result of the partial substitution of Ca^2+^ with smaller Gd^3+^ ions at both cation sites in the apatite-type structure [44]. The calculated data are in good agreement with previously reported Gd^3+^-doped hydroxyapatite [43]. Thus, it can be concluded that the impurity apatite phases are also doped by the Gd^3+^ ions. The Ca_2_P_2_O_7_ phase is not doped by Gd^3+^ ions, as the unit cell parameters do not change significantly, accounting the possible error range (Table 2). The schematic representation of substitution of Ca^2+^ by Gd^3+^ ions in apatite (Figure 3a) and whitlockite (Figure 3b) phases is shown.

### 3.3. EPR Study

For EPR studies, only the Gd-TCP-pc (i.e., the sample obtained by precipitation from aqueous solutions of salts) powder was used, as it is more phase-pure with respect to the product obtained by mechano-chemical activation, i.e., 96% of *β*-TCP phase vs. 73%, respectively. The sample with a small concentration of Gd—0.001-Gd-TCP—was chosen because higher concentrations of Gd in EPR might result in peaks’ conglomerating and broadening.

The EPR spectra were measured at 25 K. In Figure 4a (black line, powder sample), the spectrum recorded at a minimum time between π/2 and π pulses, τ = 0.18 μs, is shown. It does not contain clearly distinguished individual lines. Measurements of the decay curves of the transverse magnetization revealed two exponential processes with T_2_ = 0.19 ± 0.01 μs and 1.0 ± 0.05 μs characteristic times. To check the nature of the appearance of the two processes, the EPR spectrum was recorded for a longer time of τ = 0.54 μs between π/2 and π pulses (Figure 4a, red line). In the case of the existence of two types of centers with different relaxation times, with increasing time, a strong decrease in the spectrum component with a short relaxation time is observed compared to the background of the spectrum component with a longer time. This transforms the shape of the EPR spectrum (redistribution of intensities), which can be seen in Figure 4a as the difference between the black and red lines. The simplest mathematical processing separates the spectrum into two components with fast T_2_ = 0.19 ± 0.01 μs and slow T_2_ = 1.0 ± 0.05 μs transverse relaxation, shown in Figure 4b as blue and red lines, respectively.

Rare earth gadolinium Gd^3+^ ion with the 4f^7^ configuration in the ^8^*S*_7/2_ ground state is paramagnetic and has electron spin *S* = 7/2 with zero orbital angular momentum *L* = 0. For this spin system, the spectrum should contain 2 ∗ *S* = 7 different transitions, which give 7 lines in the EPR spectrum. The intensity of each of these lines will be proportional to the square of the matrix element $S_{x}=\sqrt{S(S+1)-M_{S}\left(1\right)M_{S}\left(2\right)}$, i.e., maximum for the transition *M*_S_ = –½ ↔ ½ and minimum for *M*_S_ = ±7/2 ↔ ±5/2. It makes it possible to compare the transitions to the lines based on their intensities, even when the spectrum is averaged by different orientations of the powder grains in the sample under study.

Center 2 has long relaxation times and an expressed structure of seven almost equidistant lines (the structure is shown by arrows in Figure 4b). The intensities of these lines increase toward the center of the spectrum, which indicates a pronounced axiality of the crystal structure corresponding to TCP crystals [45,46]. Center 1 with short relaxation times does not have such a structure and its linewidth is much higher than that of Center 2. Therefore, it can be attributed to the impurity of Gd^3+^ in the admixture phases, the low crystallinity of which leads to an additional broadening of the EPR lines.

The spin-lattice relaxation rate was measured for both centers at 25 K. The values of the magnetic field were B_0_ = 344.3 mT (contributions are made by both types of centers) and B_0_ = 118.2 mT (there is only a contribution from the center 2). The spin-lattice relaxation rate of Gd^3+^ for both centers turned out to be short, T_1_ = 30 ± 2 μs, which is comparable to the relaxation of gadolinium in other crystals [47]. The electron-nuclear interactions were measured at 12 K. For this temperature, T_1_ = 160 ± 8 μs, which is sufficient for the HYSCORE sequence. The electron-nuclear interactions were detected at two values 118.2 mT and 344.3 mT of the magnetic field induction B_0_, which allowed the observation to be made of the total spectrum HYSCORE (Figure 5a) and the spectrum only from Center 2 (Figure 5b). A strong interaction with the phosphorus nucleus was observed for Center 2 (Figure 5b). The peak corresponding to this interaction was narrow, which indicates the absence of a significant distribution in the Gd–P distances and a small distortion of the crystal structure. In the case of the sum of two HYSCORE spectra (Figure 5a), the electron-nuclear interaction with phosphorus was blurred, and it was much stronger than it should be for the transition *M*_S_ = –^1^/_2_ ↔ ^1^/_2_, which also indicates the localization of Gd^3+^ in a low crystalline phase.

As it follows from the EPR experiments, two types of paramagnetic Gd^3+^ ions are found in the investigated materials, which differ in their electronic spin–spin relaxation (dephasing) times with T_2_ = 0.19 ± 0.01 μs (type 1, fast relaxing) and T_2_ = 1.0 ± 0.05 μs (type 2, slow relaxing), correspondingly. This suggests that gadolinium is present in two different phases.

The type 1 Gd does not reveal a resolved fine structure in the EPR spectrum. It can be caused by a wide distance distribution of ions (which creates a crystal field on the Gd^3+^) in the neighborhood of the paramagnetic ion environment. In the HYSCORE spectrum, no interaction with nuclei is observed, likely also due to the distribution of the Gd^3+^-nucleus distance, causing a strong broadening of the HYSCORE peaks. It can be speculated that the type 1 center is in the amorphous phase or in the phase containing a number of dislocations.

The type 2 Gd has a resolved fine structure in the EPR spectrum and reveals a hyperfine interaction with the ^31^P nuclei in the HYSCORE experiments, meaning that the spread of the distances between the paramagnetic ion and the nearest nuclei is small (quite modest). Therefore, it can be suggested that this type of Gd is detected in a crystal phase characterized by a low number of defects.

### 3.4. Thermogravimetric Analysis

To investigate the thermal behavior of powders, 0.1Gd-TCP samples obtained by precipitation and mechano-chemical activation methods were taken, in order to better investigate the effect of Gd^3+^. The thermal behavior of the 0.1Gd-TCP-pc and 0.1Gd-TCP-ma samples differs significantly (Table 3). The thermal decomposition of 0.1Gd-TCP-pc proceeds in several steps and is accompanied by a strong endothermic effect at 250 °C, corresponding to the removal of chemically bonded water molecules. During the thermal decomposition of 0.1Gd-TCP-ma, two overlapping endo-effects are observed in the range of 80–200 °C, accompanied by a mass loss of 15%, which can be attributed to the removal of chemically coordinated water molecules.

From the results presented in Table 3, it can be seen that the 0.1Gd-TCP-ma powder is more thermally stable: in this case, the mass loss is two times lower and starts at a higher temperature. 

### 3.5. Specific Surface Area of Powders

Specific surface area experimental data for all the samples obtained by the BET method are presented in Table 4. As one can see from the table, the specific surface area of powders obtained by precipitation from aqueous solutions is more than twice higher compared to the specific surface area of powders obtained by mechano-chemical activation. The different specific surface area of the powders is related to the methods of their preparation. Powders obtained by precipitation follow the so-called “bottom up” path, i.e., individual ions interact with each other in the solution volume with the formation of a large number of crystallization centers. As a result of such crystallization, the concentration of ions in solution rapidly decreases and the growth of particles stops. This leads to the formation of nanoparticles with a size of tens of nanometers (see Section 3.6). Powders obtained by mechanochemical activation pass the so-called “top down” path, i.e., the initial oxides with certain particle sizes reduce their size as a result of grinding. The interaction takes place between these small particles and leads to the formation of larger particles compared to those obtained by the precipitation method. The incorporation of Gd^3+^ into the TCP structure leads to a decrease in the specific surface area of the powders, which is equivalent to an increase in the particle size. An increase in the particle size of the Gd-TCP-pc samples compared to TCP-ma can be associated with the formation of a smaller number of crystallization centers in the solution with gadolinium. A smaller number of crystallization centers is due to the fact that the formation of Gd-TCP-pc requires the combination of three particles (the Ca^2+^, Gd^3+^ cations, and the PO_4_^3–^ anion), whereas the formation of TCP-pc requires a combination of two particles Ca^2+^ and PO_4_^3–^.

### 3.6. TEM Studies of Powders

For TEM studies, *β*-TCP-pc, *β*-TCP-ma, 0.01Gd-TCP-pc, and 0.1Gd-TCP-pc powder samples were selected. The *β*-TCP-pc and *β*-TCP-ma samples were chosen in order to observe the particle differences prepared by precipitation (1) and mechano-chemical activation (2). As one can see from Figure 6a,b, small particles of tens of nanometers are prepared by synthesis route (1), while in the case of synthesis route (2), the particle size is in the range of hundreds of nanometers. For the Gd-TCP-pc with different substitution amounts, no significant differences are observed with respect to the pure *β*-TCP-pc sample, as expected.

### 3.7. SEM Observations of Ceramics

For SEM investigation, 0.1Gd-TCP-pc and 0.1Gd-TCP-ma ceramic samples with the highest concentration of Gd were used, in order to better observe the influence of Gd on the samples’ morphology and microstructure. The SEM images of the synthesized 0.1Gd-TCP ceramic samples after sintering at 1100 °C are presented in Figure 7. The microstructure of ceramic sample obtained from the powder synthesized by precipitation from solutions is inhomogeneous: there are both large conglomerates, consisting of small grains, and smaller grains, ranging in size from 1 up to about 50 μm (Figure 7 and Table 4). The ceramics obtained from powder synthesized using mechano-chemical activation look more uniform; the grain size varies from 0.4 to 1.4 μm (Figure 7 and Table 4). The comparison of the grain sizes of Gd-TCP, obtained using two methods, shows that larger grains and conglomerates of grains are formed in the case of using the precipitation method. This can be explained by the fact that, as a result of the synthesis by precipitation from aqueous solutions, more dispersed (nanoscale) powders are obtained (see TEM results, Section 3.6), compared to powders synthesized by mechano-chemical activation (see TEM results, Section 3.6). During the drying process, agglomeration of nanopowders occurs, accompanied by a decrease in surface energy. When removing water having a high surface tension (σ = 72.7 MJ/m^2^), aggregation of nanoparticles dispersed in an aqueous suspension occurs, accompanied by the formation of micro-scale aggregates. During further sintering at a higher temperature of 1100 °C, the particles inside the aggregate unit are tightly sintered, and, therefore, after sintering, initial nanoparticles are transformed into particles of 5–10 µm. Thus, as a result of the sintering of powders synthesized by precipitation from aqueous solutions, ceramics with a large grain size spread are obtained (Figure 7 and Table 4).

In the case of sintering of the 0.1Gd-TCP-ma powder, ceramics with a more uniform grain size are obtained. When using the 0.1Gd-TCP-pc powders, ceramics are composed by grains of dissimilar size (Figure 7a): larger grains are composed of smaller, joined grains. This is due to the fact that smaller crystallites obtained by the precipitation method from aqueous solutions agglomerate during drying, and at 1100 °C, their sintering occurs inside the agglomerates. This fact is confirmed by the size of the ceramic grains (Table 5): the average grain size of ceramics from the 0.1Gd-TCP-pc powder is 2.2 μm. It is almost three times larger than the grain size of ceramics from the 0.1Gd-TCP-ma powder. The grain size presented in Table 5 was determined from the SEM data by means of the random secant method.

The pore size, calculated from the SEM data by the random secant method for the 0.1Gd-TCP-pc ceramics (Figure 7a,b), varies from 1–2 µm up to 5–10 µm. The sintering process was carried out at 1100 °C, and the porosity at this temperature is maintained, following from the SEM data described above. In connection with this, the bending strength of ceramics is low—about 30 MPa, and the fracture curve corresponds to the brittle fracture characteristic for ceramic samples [48]. In contrast, the 0.1Gd-TCP-ma ceramics are characterized by a more homogeneous structure (see Figure 7c,d), and its bending strength of 38.6 MPa is higher, which is connected to the fact that the pore size, in this case, does not exceed 2 µm.

### 3.8. Antibacterial Activity Study

The results obtained by applying the microorganisms (*S. aureus, E. coli, S. typhimurium, E. faecalis,* and *C. albicans*) grown in the presence of 0.1Gd-TCP-pc and 0.1Gd-TCP-ma powders demonstrate a good inhibition of growth for all the microorganism species. The values of the mean OD, standard deviation, %growth, and %inhibition for each microorganism for two tested samples, after 24 h of incubation, are detailed in Table 6 and Figure 8. For all the microorganisms, there is a good inhibition of growth (a minimum of 12.2% and a maximum of 30.4%). As can be observed, the 0.1Gd-TCP-pc inhibits the growth of *S. aureus, E. coli, E. faecalis*, and *C. albicans* better than the 0.1Gd-TCP-ma. Only for the *S. typhimurium*, the inhibition of growth is slightly higher for the 0.1Gd-TCP-ma compared to the 0.1-Gd-TCP-pc. 

### 3.9. Metabolic Activity Study

The metabolic activity study of the NCTC L929 cells performed by the MTT test (Figure 9) showed the absence of significant differences between the experiment and control in the ceramic samples of pure *β*-TCP, 0.1Gd-TCP-pc, and 0.01Gd-TCP-pc obtained from powders synthesized by the precipitation method, indicating the absence of toxic effects for three-day extracts (Figure 9). The extracts from the 0.1Gd-TCP-ma sample have an inhibitory effect on the NCTC L929 fibroblasts (reliability according to the Mann–Whitney *U*-test *p* < 0.01) (see Figure 8).

However, the inoculation of cells onto the surface of the tested materials and the study of their viability (by the direct contact method) allowed us to establish the absence of toxic effects in the Gd-TCP samples. Images of cells cultured on the surface of materials are shown in Figure 10.

The DPSC test study showed that there is a decrease in the total number of cells on ceramic samples doped with Gd^3+^ ions, obtained by mechano-chemical activation synthesis, in comparison with the control (see Figure 11). However, the number of dead cells on the surface of all studied ceramic materials is insignificant, which indicates the absence of cytotoxicity of these ceramic materials. Thus, the decrease in cellular activity is not associated with cell death. Therefore, the 0.01Gd-TCP and 0.1Gd-TCP ceramic samples obtained from powders synthesized by both the precipitation and the mechano-chemical activation can be recognized as biocompatible.

## 4. Conclusions

It was established that the synthesis method influenced the phase composition of the prepared Gd-substituted TCP powders: the precipitation from solutions method resulted in a more homogeneous phase composition of *β*-TCP-type. In particular, the 0.1Gd-TCP-pc and 0.01Gd-TCP-pc powders contained 96 wt% of *β*-TCP, whereas the 0.1Gd-TCP-ma and 0.01Gd-TCP-ma contained 73 and 87 wt% of *β*-TCP, respectively, as confirmed by the XRD and FT-IR methods. Both synthesis routes, the precipitation from solutions and mechano-chemical activation, led to the effective substitution of Ca^2+^ ions by Gd^3+^ in both *β*-TCP (main phase) and apatite (admixture) phases. These results were revealed by the EPR method, which also confirmed the oxidation state of the Gd^3+^ ions. It was shown that Gd^3+^ ions were localized directly in the crystal lattice, with a small scattering of Ca–P distances and a slight distortion in the crystal structure. The presence of two types of paramagnetic Gd^3+^ ion contributions in the EPR spectra of the investigated materials was associated with the presence of gadolinium in two different phases, the main *β*-TCP phase and impurity phases (apatite/pyrophosphate), or a secondary amorphous phase, in agreement with the XRD results.

The 0.1Gd-TCP-ma powder was more thermally stable with respect to the 0.1Gd-TCP-pc: its mass loss was two times lower and started at a higher temperature. The specific surface area of the prepared powders obtained by precipitation from aqueous solutions was more than twice higher compared to the specific surface area of powders obtained by mechano-chemical activation. In particular, the specific surface area value obtained for 0.01Gd-TCP-pc was 72.4 m^2^/g, that for 0.1Gd-TCP-pc was 66.4 m^2^/g, that for 0.01Gd-TCP-ma was 35.8 m^2^/g, and that for 0.1Gd-TCP-ma was 32.7 m^2^/g.

The method of synthesis also affected the size and morphology of the obtained powders. TEM observations revealed that, in the case of the precipitation synthesis, the powders particle size was tens of nanometers, while, in the case of mechano-chemical activation, particles of hundreds of nanometers were obtained.

The SEM microstructure of ceramics obtained from the powder synthesized by precipitation was inhomogeneous, consisting of grains of 1–50 μm. The ceramics obtained from powder synthesized using mechano-chemical activation were more uniform, with grain sizes of 0.4–1.4 μm. The pore size for 0.1Gd-TCP-pc ceramics varied from 1–2 µm up to 5–10 µm with a bending strength of about 30 MPa. The 0.1Gd-TCP-ma ceramics were more homogeneous, with a pore size of 2 µm and a bending strength of about 39 MPa.

The results of the antimicrobial activity tests showed that 0.1Gd-TCP-pc and 0.1Gd-TCP-ma materials were able to inhibit the growth of the four tested bacteria (*S. aureus*, *E. coli*, *S. typhimurium*, and *E. faecalis*) and one tested fungus (*C albicans*) with respect to the relative controls: about 30% of inhibition for 0.1Gd-TCP-pc and about 28% for 0.1Gd-TCP-ma were detected. In particular, the 0.1Gd-TCP-pc inhibited *S. aureus*, *E. coli*, *E. faecalis*, and *C. albicans* more effectively than the 0.1Gd-TCP-ma. The growth of *S. typhimurium* was inhibited slightly better by the 0.1Gd-TCP-ma with respect to the 0.1Gd-TCP-pc.

According to the results of the in vitro testing, all the prepared ceramics can be recognized as biocompatible. The MTT test results on the metabolic activity of the NCTC L929 cells evidenced the absence of toxic effects. The DPSC viability study showed the absence of cytotoxicity of the prepared ceramic materials doped with Gd ions with no difference for the synthesis route.

## Figures and Tables

**Figure 1 nanomaterials-12-00852-f001:**
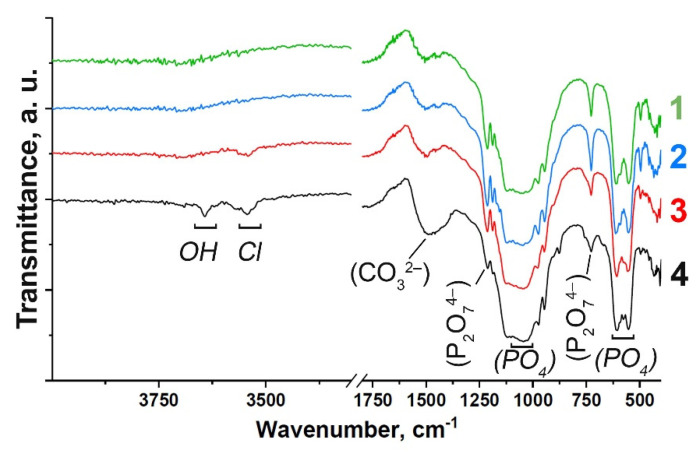
FT-IR spectra of Gd-substituted TCP powders obtained by precipitation: 0.1Gd-TCP-pc (1) and 0.01Gd-TCP-pc (2), and mechano-chemical activation: 0.1Gd-TCP-ma (3) and 0.01Gd-TCP-ma (4).

**Figure 2 nanomaterials-12-00852-f002:**
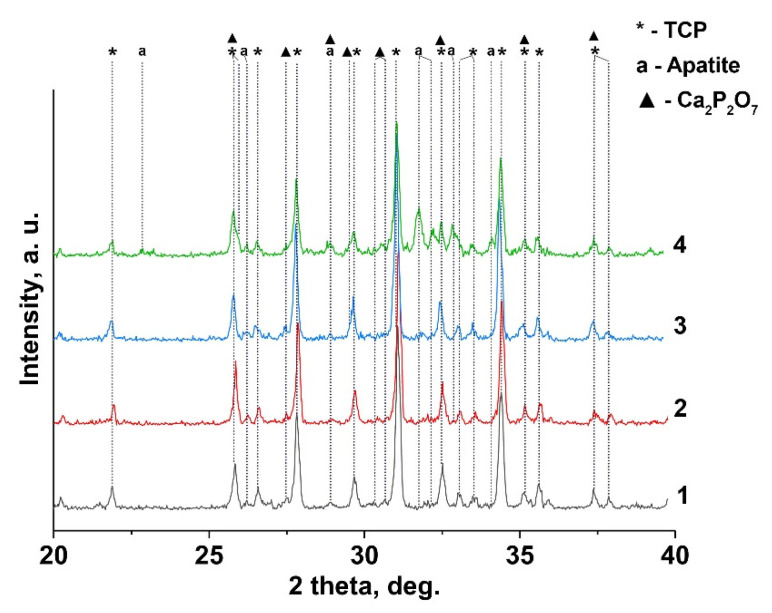
XRD pattern of Gd-TCP powders prepared by precipitation and mechano-chemical activation methods: 0.01-Gd-TCP-pc (1), 0.01-Gd-TCP-ma (2), 0.1-Gd-TCP-pc (3), and 0.1-Gd-TCP-ma (4).

**Figure 3 nanomaterials-12-00852-f003:**
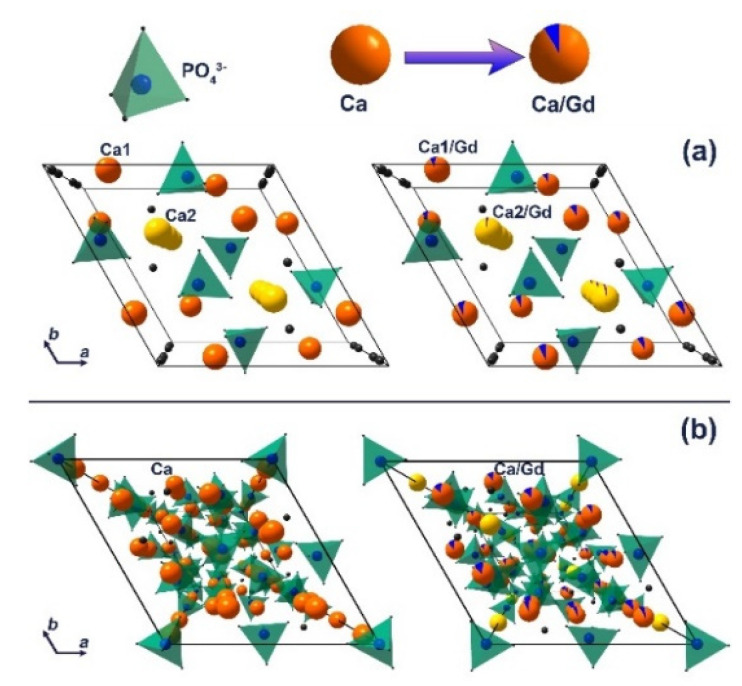
The schematic representation of substitution of Ca^2+^ by Gd^3+^ ions in apatite (**a**) and *β*-TCP (**b**) phases.

**Figure 4 nanomaterials-12-00852-f004:**
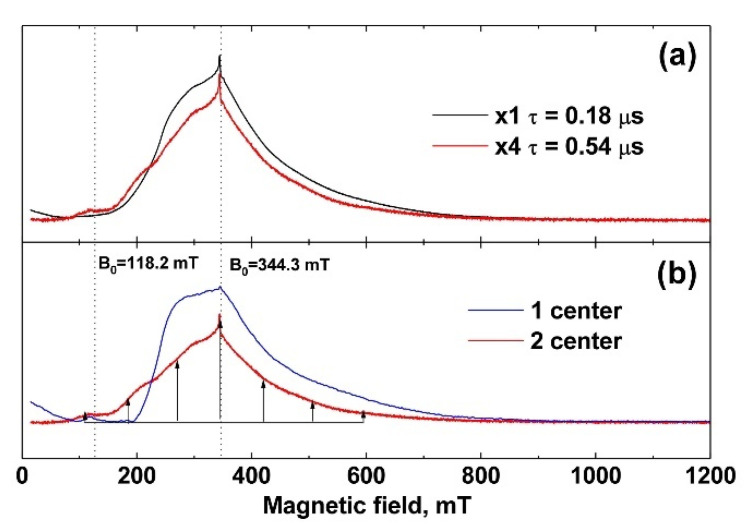
(**a**) EPR spectra of 0.001 Gd-TCP-pc. The black line corresponds to the time between pulses τ = 0.18 μs, and the red line corresponds to τ = 0.54 μs (increased in amplitude by a factor of 4); (**b**) EPR spectrum components differing in decay times T_2_ of transverse magnetization. The component indicated by the blue line corresponds to T_2_ = 0.19 ± 0.01 μs, and that indicated by the red line corresponds to T_2_ = 1.0 ± 0.05 μs.

**Figure 5 nanomaterials-12-00852-f005:**
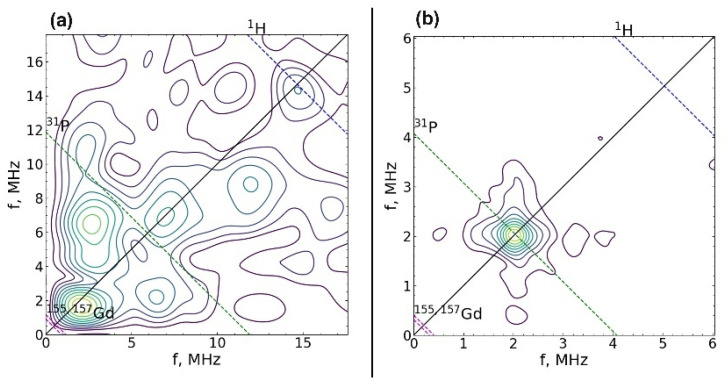
2D spectrum of electron-nuclear interactions measured using the HYSCORE sequence at magnetic field induction B_0_ = 344.3 mT (**a**) and 118.2 mT (**b**). Dashed lines show the regions of expected interactions with ^1^H, ^31^P nuclei, and ^155^Gd and ^157^Gd isotopes.

**Figure 6 nanomaterials-12-00852-f006:**
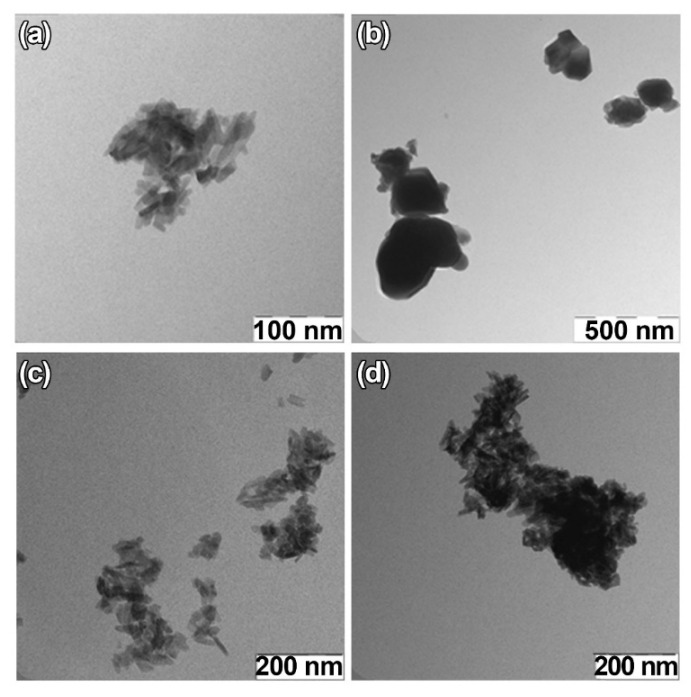
TEM images of TCP-pc (**a**), TCP-ma (**b**), 0.01Gd-TCP-pc (**c**), and 0.1Gd-TCP-pc (**d**).

**Figure 7 nanomaterials-12-00852-f007:**
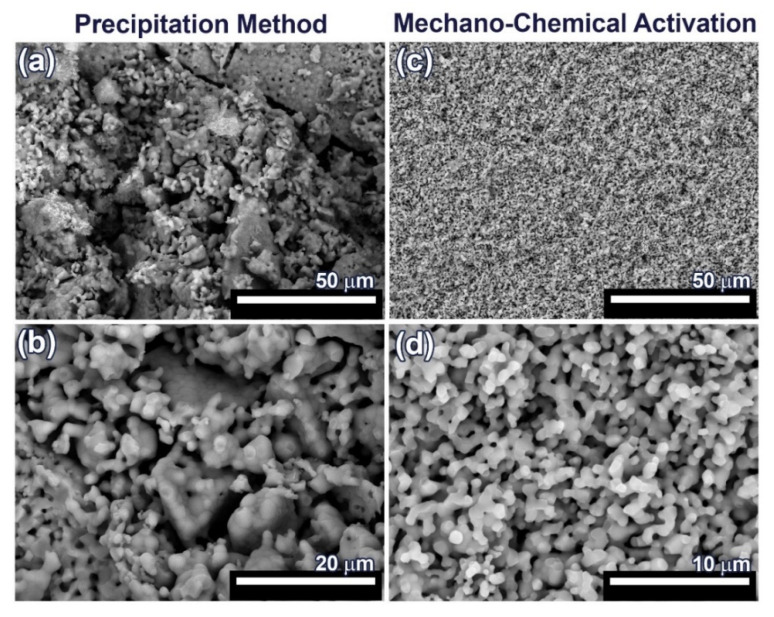
SEM images of 0.1Gd-TCP-pc (**a**,**b**) and 0.1Gd-TCP-ma (**c**,**d**) followed by sintering at 1100 °C.

**Figure 8 nanomaterials-12-00852-f008:**
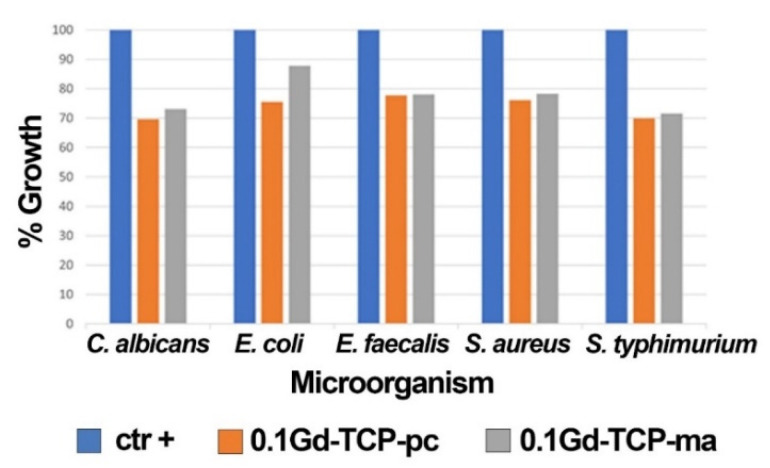
Results of the growth rate of *S. aureus*, *E. coli*, *S. typhimurium*, *E. faecalis*, and *C. albicans* in the presence and absence (positive control, ctr +) of 0.1Gd-TCP-pc and 0.1Gd-TCP-ma. The %growth was calculated from three independent experiments.

**Figure 9 nanomaterials-12-00852-f009:**
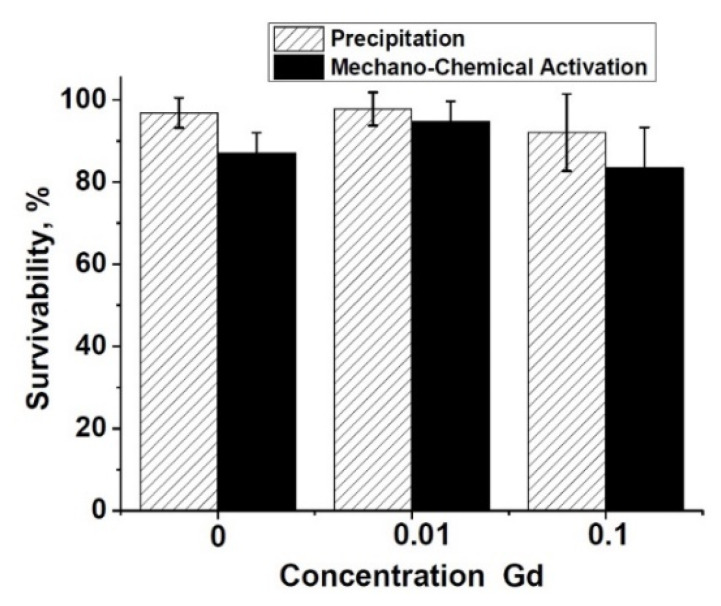
Metabolic activity of the NCTC L929 cells after 24 h of incubation with 3 day extracts of pure *β*-TCP (1), 0.01Gd-TCP (2), and 0.1Gd-TCP (3) synthesized by precipitation and mechano-chemical activation.

**Figure 10 nanomaterials-12-00852-f010:**
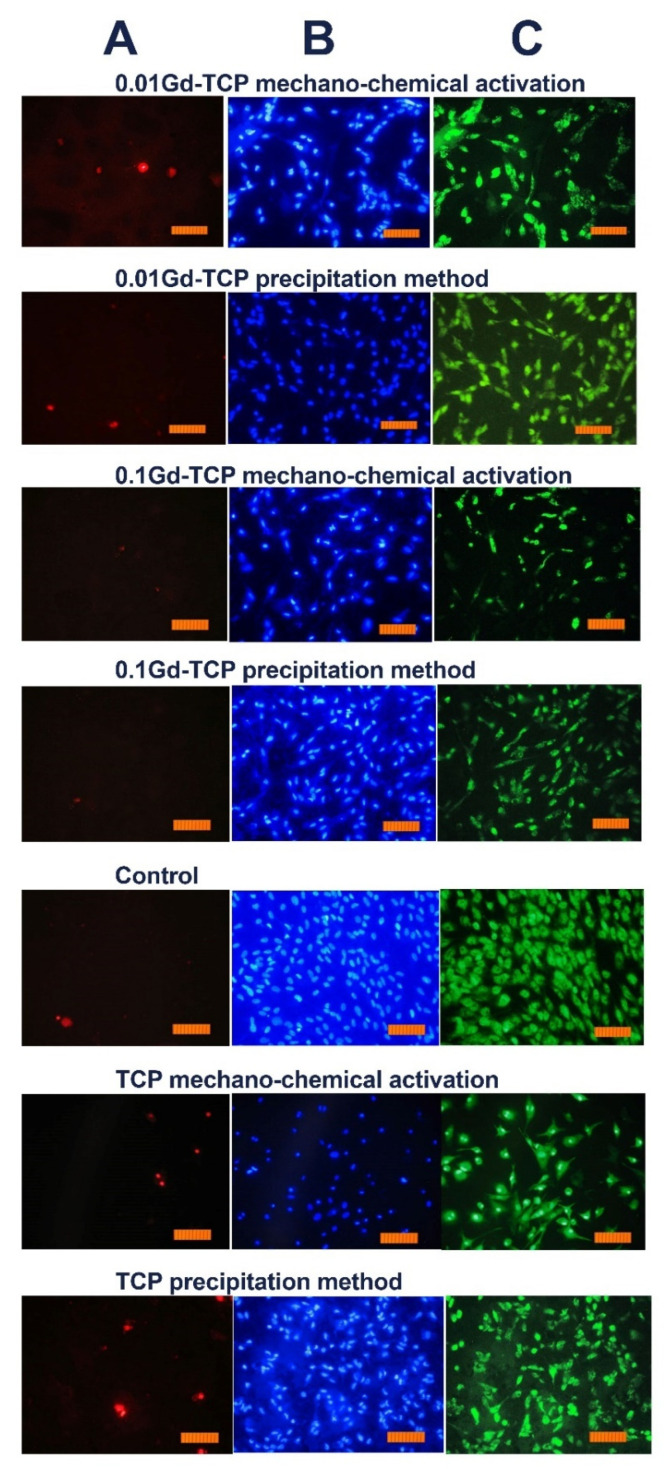
Images of the DPSC cells during incubation on the surface of tested ceramic materials during the first day after inoculation. Dyeing propidium iodide (**A**), Hoechst 33,342 (**B**), and SYTO 9 (**C**) (scale 100 µm).

**Figure 11 nanomaterials-12-00852-f011:**
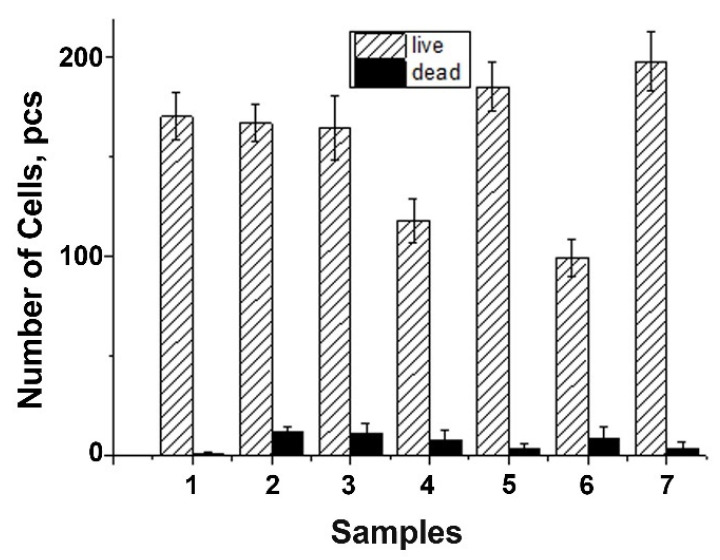
Number of DPSC cells seeded on the surface of ceramics: TCP-pc (1), TCP-ma (2), 0.01Gd-TCP-pc (3), 0.01Gd-TCP-ma (4), 0.1Gd-TCP-pc (5), 0.1Gd-TCP-ma (6), and control sample (7). The error bars show the standard deviation.

**Table 1 nanomaterials-12-00852-t001:** Vibration modes in the FT-IR spectrum of Gd-substituted TCP powders.

Assignment	Wavenumber, cm^–1^	Vibration Mode	Sample
OH^–^(Cl^−^)	3644, 3544	Valence	(4); weak: (3)
PO_4_^3–^	973	ν_1_	(1), (2), (3), (4)
PO_4_^3–^	493, 432	ν_2_	(1), (2); weak: (3) (4)
PO_4_^3–^	1123, 1044	ν_3_	(1), (2), (3), (4)
PO_4_^3–^	607, 574, 556	ν_4_	(1), (2), (3), (4)
P_2_O_7_^4–^	1212, 727	Valence P-O-P	(1), (2); weak: (3), (4)
CO_3_^2–^	1490	ν_3_	(1), (2), (3), (4)
CO_3_^2–^	875	ν_2_	(1), (2), (3), (4)

**Table 3 nanomaterials-12-00852-t003:** Thermogravimetric analysis data for 0.1Gd-TCP samples obtained by precipitation and mechano-chemical activation methods.

Sample	Temperature, °C	Thermal Effect Type	Mass Loss, wt%	Mass Loss Corresponds to
0.1Gd-TCP-pc	80	endothermic	5	physically adsorbed H_2_O
	250	endothermic	45	chemically coordinated H_2_O
0.1Gd-TCP-ma	110–250	endothermic	20	two-stage H_2_O elimination

**Table 4 nanomaterials-12-00852-t004:** BET data for TCP and Gd-TCP, synthesized by precipitations and mechano-activation methods.

N	Sample	Surface Area, m^2^/g
1	TCP-pc	77.07 ± 0.09
2	0.01Gd-TCP-pc	72.49 ± 0.07
3	0.1Gd-TCP-pc	66.41 ± 0.13
4	TCP-ma	30.09 ± 0.01
5	0.01Gd-TCP-ma	35.88 ± 0.04
6	0.1Gd-TCP-ma	32.73 ± 0.03

**Table 5 nanomaterials-12-00852-t005:** Average grain size of 0.1Gd-TCP ceramics.

Grain Size, μm	0.1Gd-TCP-pc	0.1Gd-TCP-ma
minimal	1.2	0.4
maximal	49.1	1.4
median	2.2	0.8

**Table 6 nanomaterials-12-00852-t006:** Mean values of OD_600_, SD, and percentage of growth and inhibition of the different microorganisms by 0.1Gd-TCP-pc and 0.1Gd-TCP-ma powders, derived from three independent experiments.

*C. albicans*				
	OD_600_	SD	% Growth	% Inhibition
ctr + (control sample)	0.711	0.008	100	0
0.1Gd-TCP-pc	0.495	0.020	69.7	30.3
0.1Gd-TCP-ma	0.519	0.007	73.1	26.9
*E. coli*				
	OD_600_	SD	% Growth	% Inhibition
ctr + (control sample)	0.845	0.025	100	0
0.1Gd-TCP-pc	0.638	0.054	75.5	24.5
0.1Gd-TCP-ma	0.741	0.015	87.8	12.2
*E. faecalis*				
	OD_600_	SD	% Growth	% Inhibition
ctr + (control sample)	0.570	0.021	100	0
0.1Gd-TCP-pc	0.443	0.011	77.8	22.2
0.1Gd-TCP-ma	0.445	0.032	78.1	21.9
*S. aureus*				
	OD_600_	SD	% Growth	% Inhibition
ctr + (control sample)	0.951	0.021	100	0.
0.1Gd-TCP-pc	0.723	0.072	76.1	23.9
0.1Gd-TCP-ma	0.744	0.016	78.3	21.7
*S. typhimurium*				
	OD_600_	SD	% Growth	% Inhibition
ctr + (control sample)	0.906	0.076	100	0
0.1Gd-TCP-pc	0.634	0.040	70.0	30.0
0.1Gd-TCP-ma	0.648	0.043	71.6	28.4

## Data Availability

The data are available upon an official and reasonable request.

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
