# Peer review of "Influence of Synthesis Conditions on Gadolinium-Substituted Tricalcium Phosphate Ceramics and Its Physicochemical, Biological, and Antibacterial Properties"

_nanomaterials, 2022, doi:10.3390/nano12050852_

Round 1

Reviewer 1 Report

Refereree prport on manuscript “The influence of synthesis conditions on gadolinium-substituted tricalcium phosphate ceramics and its physico-chemical, biological and antibacterial properties”

Overall, the scientific quality of the submitted manuscript is fine. The discussions based on the experimental results are sound. Several minor concerns are proposed herein for revision.

  1. First question concerns Figure 1and table 1. The 4 curves shown in the figure have some differences, while the table shows only one column. For clarity, please give a detailed description of each spectrum in Table 1.
  2. The second question is also about Table 1 and Fih.1. Is there a dependence of the spectra on the particle size? For phosphates, this has been described in detail by: Savchyn, P.; Karbovnyk, I.; Vistovskyy, V.; Voloshinovskii, A.; Pankratov, V.; Cestelli Guidi, M.; Mirri, C.; Myahkota, O.; Riabtseva, A.; Mitina, N. Vibrational properties of LaPO4nanoparticles in mid-and far-infrared domain.  Appl. Phys. 2012112, 124309.
  3. In the case of the substitution of Ca2+ ions by Gd3+ , is there any charge compensator?

Author Response

Overall, the scientific quality of the submitted manuscript is fine. The discussions based on the experimental results are sound. Several minor concerns are proposed herein for revision.

  1. First question concerns Figure 1and table 1. The 4 curves shown in the figure have some differences, while the table shows only one column. For clarity, please give a detailed description of each spectrum in Table 1.

REPLY: We thank the Reviewer for the valuable comment and suggestion. As suggested, Table1 was corrected and was complemented by relevant description. 

2. The second question is also about Table 1 and Fih.1. Is there a dependence of the spectra on the particle size? For phosphates, this has been described in detail by: Savchyn, P.; Karbovnyk, I.; Vistovskyy, V.; Voloshinovskii, A.; Pankratov, V.; Cestelli Guidi, M.; Mirri, C.; Myahkota, O.; Riabtseva, A.; Mitina, N. Vibrational properties of LaPO4nanoparticles in mid-and far-infrared domain.  Appl. Phys. 2012112, 124309.

REPLY: We believe that in this case it is hardly possible to isolate the influence of the size factor on the shape and character of the IR spectra. However, the corresponding paragraph with a discussion of the role of the size factor and reference to the proposed publication was added to the text of the article in section 3.1.The new proposed suitable reference has the number [36].

3. In the case of the substitution of Ca2+ ions by Gd3+ , is there any charge compensator?

REPLY: The following sentence was added in the revised version of the manuscript on page 7: “The scheme of the substitution is 3Ca2+ → 2R3+ +â–¡ (â–¡-vacancy) with the formation of vacancy and no additional charge compensators are required.

Reviewer 2 Report

My comments are in attached file.

Author Response

 The general level of this study is high and manuscript could be considered for publication after minor revision reasonable to increase the paper quality. My several corrections proposed for the text are listed below for author consideration.

Reply: We thank the Reviewer for careful reading and useful corrections. All the corrections proposed by the Reviewer were done in the revised version of the manuscript.

Page 1

Julietta V. Rau 9,7*

Julietta V. Rau 7,9*

Reply: The main affiliation of J. Rau is N9, and, therefore, it is indicated as the first one.

Page 1

In the case of the synthesis route (1), the powders with particle size of tens of nm were obtained, while in the case of the synthesis (2), particle size was hundreds of nm, revealed by Transmission Electron Microscopy.

In the case of the synthesis route (1), the powders with particle size of tens of nm were obtained, while in the case of the synthesis (2), particle size was hundreds of nm, as revealed by Transmission Electron Microscopy.

Reply: The suggested correction has been made.

Page 2 

Osteopenia is associated with several outcomes including functional decline, high risk of fractures and frailty.

Osteopenia is associated with several outcomes, including functional decline, high risk of fractures and frailty.

Reply: The suggested correction has been made.

Page 2

Antibacterial characteristics can be imparted by Cu2+, Fe3+, Zn2+ and Ag+ ions [12,14–16], biocompatibility properties and, namely, cell proliferation on the materials surface can be improved by Fe3+ and Sr2+ ions substitution [5,14,17–19].

Antibacterial characteristics can be imparted by Cu2+, Fe3+, Zn2+ and Ag+ ions [12,14–16], and biocompatibility properties and, namely, cell proliferation on the materials surface can be improved by Fe3+ and Sr2+ ions substitution [5,14,17–19].

Reply: The suggested correction has been made.

Page 3

The prepared powder samples were characterized by Fourier-transform infrared spectroscopy (FT-IR), X-Ray diffraction (XRD), electron paramagnetic resonance spectroscopy (EPR), thermogravimetric analysis (TGA), specific surface area, scanning electrons microscopy (SEM), and Transmission electron microscopy (TEM) methods.

The prepared powder samples were characterized by the Fourier-transform infrared spectroscopy (FT-IR), X-Ray diffraction (XRD), electron paramagnetic resonance spectroscopy (EPR), thermogravimetric analysis (TGA), specific surface area, scanning electrons microscopy (SEM), and Transmission electron microscopy (TEM) methods.

Reply: The suggested correction has been made.

Page 3

The fibroblast cell line NCTC clone L929 was used for cytotoxicity studies, the viability of postnatal human dental pulp stem cells (DPSCs) was investigated.

The fibroblast cell line NCTC clone L929 was used for cytotoxicity studies, and the viability of postnatal human dental pulp stem cells (DPSCs) was investigated.

Reply: The suggested correction has been made.

Page 3

The powders obtained by the synthesis methods (1) and (2) were heat-treated at 900 °Ð¡ for 1 hour.

The powders obtained by the synthesis methods (1) and (2) were heat-treated at 900 °Ð¡ for 1 h. 3

Reply: The suggested correction has been made.

Page 4

Powder X-Ray diffraction patterns were collected on the Rigaku D/MAX 2500 (Ni-radiation filter Cu Kα, θ/2θ geometry).

Powder X-Ray diffraction patterns were collected on a Rigaku D/MAX 2500 (Ni-radiation filter Cu Kα, θ/2θ geometry) diffractometer.

Reply: The suggested correction has been made.

Page 4

Crystallographic data of space group (SG), unit cell and atomic coordinates 159 of β-Ca3(PO4)2 (PDF#4 No 70-2065), Ca10(PO4)6(OH)2 (ICDD 183744) and Ca2P2O7 (PDF#4 160 No 04-009-3876) were applied as initial parameters.

Besides card numbers, the related original papers should be cited.

Reply: The suggested correction has been made, and the card numbers are cited on page 4.

Page 4

The unit cell parameters were refined, the atomic coordinates were taken without refinement.

The unit cell parameters were refined, but the atomic coordinates were taken without refinement.

Reply: The suggested correction has been made.

Page 4

Electron paramagnetic resonance spectra in a pulse mode were recorded using Bruker Elexsys E580 spectrometer in the X-band microwave range (νMW = 9.61 GHz) at the temperature of 12 K and 25 K.

Electron paramagnetic resonance spectra in a pulse mode were recorded using a Bruker Elexsys E580 spectrometer in the X-band microwave range (νMW = 9.61 GHz) at the temperature of 12 and 25 K.

Reply: The suggested correction has been made.

Page 4

Gadolinium-substituted TCP - 0.001Gd-TCP - with a gadolinium content of 0.0067 wt. % was synthesized for the EPR spectroscopic studies by precipitation method.

Gadolinium-substituted TCP - 0.001Gd-TCP - with a gadolinium content of 0.0067 wt. % was synthesized for the EPR spectroscopic studies by the precipitation method.

Reply: The suggested correction has been made.

Page 4 

Scanning electron microscopy observations of Gd-substituted TCP were performed to study the microstructure of the ceramic materials by the Tescan Vega II scanning electron microscope.

Scanning electron microscopy observations of Gd-substituted TCP were performed to study the microstructure of the ceramic materials by a Tescan Vega II scanning electron microscope.

Reply: The suggested correction has been made.

Page 4

Transmission electron microscopy images of pure -TCP-pc, -TCP-ma and Gd-substituted TCP – 0.1Gd-TCP-pc and 0.01Gd-TCP-pc –were obtained by using the LEO 912 AB OMEGA microscope (Carl Zeiss, Oberckochen, Germany).

Transmission electron microscopy images of pure -TCP-pc, -TCP-ma and Gd-substituted TCP – 0.1Gd-TCP-pc and 0.01Gd-TCP-pc –were obtained by using a LEO 912 AB OMEGA microscope (Carl Zeiss, Oberckochen, Germany).

Reply: The suggested correction has been made.

Page 5

It should also be noted the presence of bands assigned to the carbonate group, which indicates partial replacement of the groups PO43– → CO32–.

The presence of bands assigned to the carbonate group should also be noted, and it indicates partial replacement of the groups PO43– → CO32–.

Reply: The suggested correction has been made.

Page 7

According to the substitution scheme 3Ca2+ → 2R3+ +â–¡ (â–¡-vacancy) and the values of the ionic radii of calcium (rVI = 1.00 Å, rVIII = 1.12 Å) and gadolinium (rVI = 0.94 Å, rVIII = 1.05 Å) [34] ions this substitution should have an impact on the unit cell parameters and volume.

According to the substitution scheme 3Ca2+ → 2R3+ +â–¡ (â–¡-vacancy) and the values of the ionic radii of calcium (rVI = 1.00 Å, rVIII = 1.12 Å) and gadolinium (rVI = 0.94 Å, rVIII = 1.05 Å) [34] ions, this substitution should have an impact on the unit cell parameters and volume.

Reply: The suggested correction has been made.

Page 8

The Ca2P2O7 phase wasn’t doped by Gd3+ ions since the unit cell parameters did not change significantly within the calculation error (Table 2). 5

The Ca2P2O7 phase wasn’t doped by Gd3+ ions since the unit cell parameters did not change significantly, accounting the possible error range (Table 2).

Reply: The suggested correction has been made.

Page 12

For TEM studies, β-TCP-pc, β-TCP-ma, 0.01Gd-TCP-pc and 0.1Gd-TCP-pc powder 436 samples were applied.

For TEM studies, β-TCP-pc, β-TCP-ma, 0.01Gd-TCP-pc and 0.1Gd-TCP-pc powder 436 samples were selected.

Reply: The suggested correction has been made.

Page 13

This can be explained by the fact that as a result of the synthesis by precipitation from aqueous solutions, more dispersed (nanoscale) powders were obtained (see TEM results, section 3.6), compared to powders synthesized by mechano-chemical activation (see TEM results, section 3.6).

This can be explained by the fact that, as a result of the synthesis by precipitation from aqueous solutions, more dispersed (nanoscale) powders were obtained (see TEM results, section 3.6), compared to powders synthesized by mechano-chemical activation (see TEM results, section 3.6).

Reply: The suggested correction has been made.

Page 14

At 1100 °C sintering process was carried out, and the porosity at this temperature was maintained as follows from the SEM data described above.

The sintering process was carried out at 1100 °C, and the porosity at this temperature was maintained, as follows from the SEM data described above.

Reply: The suggested correction has been made.

Page 14

Whereas the 0.1Gd-TCP-ma ceramics were characterized by a more homogeneous structure (see Figure 7 (c,d)), and its bending strength – 38.6 MPa was higher, which was connected to the fact the pore size, in this case, did not exceed 2 μm.

Whereas the 0.1Gd-TCP-ma ceramics were characterized by a more homogeneous structure (see Figure 7 (c,d)), and its bending strength – 38.6 MPa was higher, which was connected to the fact that the pore size, in this case, did not exceed 2 μm.

Reply: The suggested correction has been made.

Page 15 6

The extracts from 0.1Gd-TCP-ma sample, have an inhibitory effect on the NCTC L929 fibroblasts (reliability according to the Mann – Whitney U-test p < 0.01) (see Figure 8).

The extracts from 0.1Gd-TCP-ma sample have an inhibitory effect on the NCTC L929 fibroblasts (reliability according to the Mann – Whitney U-test p < 0.01) (see Figure 8).

Reply: The suggested correction has been made.

Page 18

In particular, the 0.1Gd-TCP-pc and 0.01Gd-TCP-pc powders contain 96 wt% of β-TCP, whereas the 0.1Gd-TCP-ma and 0.01Gd-TCP-ma – 73 and 87 wt% of β-TCP, respectively, confirmed by the XRD and FT-IR methods.

In particular, the 0.1Gd-TCP-pc and 0.01Gd-TCP-pc powders contain 96 wt% of β-TCP, whereas the 0.1Gd-TCP-ma and 0.01Gd-TCP-ma – 73 and 87 wt% of β-TCP, respectively, as confirmed by the XRD and FT-IR methods.

Reply: The suggested correction has been made

Page 19

TEM observations revealed that in the case of the precipitation synthesis, the powders particle size was tens of nm, while in the case of mechano-chemical activation, hundreds of nm particles were obtained.

TEM observations revealed that, in the case of the precipitation synthesis, the powders particle size was tens of nm, while, in the case of mechano-chemical activation, hundreds of nm particles were obtained.

Reply: The suggested correction has been made

Round 2

Reviewer 1 Report

The article has been significantly improved and can be recommended for publication.